# Health-zkIDM: A Healthcare Identity System Based on Fabric Blockchain and Zero-Knowledge Proof

**DOI:** 10.3390/s22207716

**Published:** 2022-10-11

**Authors:** Tianyu Bai, Yangsheng Hu, Jianfeng He, Hongbo Fan, Zhenzhou An

**Affiliations:** 1Faculty of Information Engineering and Automation, Kunming University of Science and Technology, Kunming 650500, China; 2Yunnan Key Laboratory of Smart City in Cyberspace Security, Kunming 650500, China

**Keywords:** decentralized identity, healthcare, zero-knowledge proof, blockchain, authentication, chaincode, hyperledger fabric

## Abstract

The issue of identity authentication for online medical services has been one of the key focuses of the healthcare industry in recent years. Most healthcare organizations use centralized identity management systems (IDMs), which not only limit the interoperability of patient identities between institutions of healthcare, but also create isolation between data islands. The more important matter is that centralized IDMs may lead to privacy disclosure. Therefore, we propose Health-zkIDM, a decentralized identity authentication system based on zero-knowledge proof and blockchain technology, which allows patients to identify and verify their identities transparently and safely in different health fields and promotes the interaction between IDM providers and patients. The users in Health-zkIDM are uniquely identified by one ID registered. The zero-knowledge proof technology is deployed on the client, which provides the user with a proof of identity information and automatically verifies the user’s identity after registration. We implemented chaincodes on the Fabric, including the upload of proof of identity information, identification, and verification functions. The experiences show that the performance of the Health-zkIDM system can achieve throughputs higher than 400 TPS in Caliper.

## 1. Introduction

With the graduation of healthcare to digitization, especially with the impact of COVID-19 in recent years, a large number of medical information services are being connected. Currently, most healthcare organizations and platforms are centralized [1], suffering from problems, such as single point of failure [2], malicious tampering [3], and internal attacks [4], which lead to problems with the confidentiality and security of patient’s private data. In addition, a single patient’s identity data may be distributed in multiple healthcare service institutions, and the information cannot be interoperable between them. The identity authentication management of patients is isolated [5], i.e., patients need to register different digital identities between institutions [6]. Thus, when patients visit the institution, some operations, such as information recovery and repeated registration, are inevitable because they forget the account.

The distributed ledger making use of blockchain technology is often decentralized, and its transaction process is tamper-resistant and transparent [7,8,9,10]. A trusted distributed authentication system can be created in an untrusted environment, allowing users to manage their own identities, thus solving most problems in the traditional centralized identity management system [11]. The decentralized feature allows anyone to manage the distributed ledger and permanently record transactions, while the tamper-resistant and transparent features prevent the modification and forgery of medical data. However, the transparent nature of blockchain technology makes it a drawback in the medical scenario [12], where sensitive information about patients cannot be stored in the blockchain directly.

Zero-knowledge proof (ZKP) is an authentication protocol that can prove that a statement is correct without revealing any information other than the statement. It can solve the transparency problem of sensitive information on the blockchain [13,14]. For the identity management system, zero-knowledge proof generates patient identification information through cryptographic practices. The information is only verified, but not decrypted and viewed during authentication, which ensures the security of patient privacy. In this paper, we propose a partially decentralized identity authentication system based on zero-knowledge proof and blockchain for healthcare scenarios. The contributions are as follows:(1)We implement a healthcare zero-knowledge identity authentication system called Health-zkIDM, which consists of four entities, namely the user of the system, healthcare provider, blockchain, and trusted third party. The user, who accesses the proposed system, is a collection of patients and healthcare providers. The user can generate identity proof information and upload it to the blockchain. Healthcare providers can verify the user’s identity by querying the blockchain for information. The trusted third party will establish the blockchain, deploy the chaincode, and provide the trusted setup for ZKP;(2)We add the zk-SNARK, a ZKP algorithm, to the Health-zkIDM system for generating the user’s identity proof information and for verifying the proof information stored in the blockchain. The real information of users is not involved in blockchain, which ensures the security of users’ privacy;(3)Health-zkIDM is implemented on the Fabric [15] consortium blockchain. The chaincode is deployed on the Fabric and is used to upload and verify the user’s identity information. The invocation of the chaincode is triggered by any healthcare provider when verifying a user’s identity;(4)We test the performance of the Health-zkIDM system using Caliper [16]. The results show that our system can reach a throughput of over 400 TPS.

The rest of this paper is organized as follows: Section 2 describes the related work of the Health-zkIDM system; in Section 3, we introduce the methodology and propose the architecture of the Health-zkIDM; Section 4 analyzes the performance and security of the system; finally, we summarize the Health-zkIDM and discuss the future work in Section 5.

## 2. Related Work

Identity management (IDM) is used to control entity identities to ensure that legitimate entities are granted access to relevant resources. IDM schemes are divided into two categories based on whether they are decentralized or not. For centralized IDM, the identity providers (IdPs) have complete centralized control over user identity management and provide authentication services. A user in a specific trusted domain cannot authenticate himself to a user in another domain. Another is federated IDM, a system that allows users in one domain to authenticate and access services in other domains, such as single-sign-on systems like Facebook Connect [17]. However, both of these approaches to IDM are often housed in a centralized server, neither of which allows full control by the user. Therefore, there are user identity security issues, which will inevitably lead to user data disclosure if the server is attacked.

Distributed ledger using blockchain technology gives researchers a new perspective on decentralized IDM. Users’ identities are no longer under the control of a single institution, thus reducing identity privacy breaches. Decentralized IDM is often divided into user-centric self-sovereign identity and decentralized trusted identity. Users under the self-sovereign identity model have full control over their identity and information [18]. The common model of self-sovereign identity includes Sovrin [19,20], uPort [21], Blockstack [22], etc. Sovrin is a global public blockchain based on Hyperledger, where only trusted institutions (banks, governments, universities, etc.) can participate in the consensus and manage blockchain, which features governance, scalability, and accessibility [23]. Sovrin enables privacy designs, such as using pseudonyms to hide customer information or employing zero-knowledge proof encryption to selectively guarantee privacy. uPort is a secure, self-controlled decentralized identity framework deployed in Ethereum that maintains users’ identities through Ethereum account addresses and wallet management. Users interact with other smart contracts using the private keys of the controlling agent contract, allowing off-chain storage of attributes linked to the uPort identity. Blockstack uses the Bitcoin blockchain to create a decentralized public key infrastructure (PKI) on top of the Namecoin network, aiming to build a decentralized new Internet infrastructure architecture. The main limitation of the self-sovereign identity approach is that the identity information is provided by the users themselves. There is no method to verify the authenticity of this identity information. In contrast to the self-sovereign identity model, a decentralized trusted identity is provided by a centralized service. The service verifies the users’ identities based on trusted credentials, such as ID cards, passports, etc. ShoCard [24] provides trusted identity verification as a representative of the decentralized trusted identity approach. It provides multi-factor authentication by binding together user identifiers, existing trusted credentials, and other identity attributes through a cryptographic hash stored in Bitcoin transactions. However, this decentralized trusted identity scheme still relies on a centralized server. Without a centralized server, users cannot authenticate to a third party. Blockchain-based identity authentication methods have different functions in different scenarios. Kuperberg et al. [25] conducted a survey on blockchain-based authentication from an enterprise and ecosystem perspective. They point out the contribution of authentication methods in enterprise scenarios. Sousa et al. [26] use blockchain for identity management in the Internet of Things (IoT) environment. They designed the system based on Sovrin, aiming to define the identity of the device and design new security mechanisms.

Our focus is on blockchain-based authentication methods in healthcare scenarios. Currently, identity verification services in healthcare are also being explored with some distributed confidentiality, for example, how to protect identity data from theft in healthcare organizations, whether service providers are following security and privacy regulations, etc. However, most identity management solutions in healthcare scenarios still rely on centralized IDM. Azaria et al. [27] first released a fully functional prototype of a blockchain system, called “MedRec”, for managing medical records. HealthChain, built by Xu et al. [28], aims to remotely monitor health data and enable the security and sharing of data in smart healthcare. Yazdinejad et al. [29] are very useful solutions for distributed applications of EHRs based on blockchain technology for cross-platform authentication when implementing secure communication among hospital networks. Based on the prototype of MedRec, Jabbar et al. [30] proposed BiiMED, which enhances data interoperability and integrity regarding sharing of EHR through the use of cross-blockchain technology. Abbas et al. [31] designed a BSDMF framework that provides secure data management between personal servers and implantable medical devices, and between cloud servers and personal servers. Wang et al. [32] used the Fabric framework, aiming to address the privacy protection problem in data sharing in healthcare scenarios. These solutions provide a very useful exploration of distributed applications of EHR. However, their significant drawback is that they do not go further to propose a decentralized IDM. The system proposed by Javed et al. [33] implements authentication management of healthcare providers and patients by healthcare regulators. Table 1 summarizes the differences and main contributions of the references. However, these methods are too cumbersome in the process of interaction among entities, which significantly increases the time overhead. Moreover, once attacked during interactive access, attackers can intercept public information on the chain, causing privacy leakage problems. Therefore, this paper aims to provide a more secure and faster decentralized IDM solution for healthcare scenarios. By invoking the chaincode, the healthcare provider can authenticate patients. This approach not only allows users to have autonomy over their identity, but a trusted third-party consortium consisting of public security, government, and healthcare regulators can also securely verify the identities of users.

## 3. Methodology

In this section, we will review some of the technical preparations required in this paper.

### 3.1. Review

#### 3.1.1. Blockchain and Smart Contracts

Since the rise of Bitcoin [34], blockchain technology has become widely known as a core technology. Distributed ledgers using blockchain technology have the characteristics of being tamper-resistant, and some of them often have the characteristics of being decentralized. According to the way the blockchain system controls the joining of nodes, blockchain can be divided into permissionless blockchain and permissioned blockchain. Blockchain can also be divided into the public blockchain, consortium blockchain, and private blockchain according to the degree of decentralization and the level of exposure of ledger information to the public. Public blockchain allows any peer to join the blockchain network. Since the public blockchain is very large, there is no user access restriction, which consumes a lot of energy. Furthermore, since anyone can read the information on the chain, public blockchain is not suitable for achieving patient privacy and confidentiality. Private blockchain still follows centralized thinking, has strict access control authority, and information is not public, so they also face the problem of centralization. Consortium blockchain is between public and private blockchain, jointly managed by several organizations. It is only open to members of a specific group and limited third parties, with characteristics of partial decentralization, strong controllability, and faster transaction speed, that can be used in many scenarios. In this paper, we adopted the most popular Hyperleger Fabric [15], hosted by the Linux Foundation, as the consortium blockchain of the proposed architecture. Hyperleger Fabric is a modular and scalable consortium blockchain platform. Smart contracts are one of the core technologies of the blockchain (called chaincode in Fabric). As distributed programs run on the blockchain, smart contracts are triggered by various events generated by various operations. As the underlying platform for blockchain application development, Hyperledger Fabric supports the implementation of permissioned blockchain and is designed to satisfy most enterprise-level requirements.

#### 3.1.2. Cryptographic Commitment and Zero-Knowledge Proofs

The cryptographic commitment scheme [35] allows one to commit the secret without revealing the secret itself, and can also verify the secret later. A cryptographic commitment scheme is a pair of probabilistic polynomial-time algorithms $\left(SGen,Com\right)$.  SGen is an initialization setup algorithm, which generates a commitment key $\;K\leftarrow SGen\left(\lambda \right)$ through a security parameter  λ. $Com_{K}\;$ is a commitment algorithm from the message space $\;M_{K}\;$ and random space $\;R_{K}\;$ mapped to the commitment space $\;C_{K}$. For a secret value $\;m\in M_{K}$, we can pick the random number $\;r\in R_{K}\;$ that computes the commitment $\;c=Com_{K}\;\left(m,r\right)\;\in \;C_{K}$. The commitment scheme has the following two properties:

(1)Hiding. Commitment value  c  will not reveal any information about the secret value  m  of any information.

(2)Binding. Given a secret value  m  and calculating its commitment value $\;c=Com_{K}\;\left(m,r\right)$, it is impossible to exist a commitment value $c^{\prime }=Com_{K}\;\left(m^{\prime },r\right)$ corresponding to another secret value $m^{\prime }$ equal to commitment c. This property ensures that different secrets cannot generate the same commitment. 

Pedersen commitment is a type of cryptographic commitment that ensures security based on the discrete logarithm assumption. Given a prime order  p  of the group  ℤp, a cyclic group  G. Selecting MK, RK=ℤp, and CK=G. Commitments are generated through the following two stages:

(1)Initialization.  SGen:g,h←G, where  g,h  is the cyclic group  G  of the generating elements.

(2)Calculation of commitment. $c=Com_{K}=g^{m}h^{r}$.

Zero Knowledge Proof [36] (ZKP) is an encryption method proposed by Goldwasser et al., in 1989. In the ZKP protocol, one of the parties (called provers) can prove to the other party (called verifiers) that they know a secret  m  through a cryptographic commitment scheme without passing any information other than the fact that they know the secret  m. A zero-knowledge set membership (ZKSM) proof enables a prover to prove that a secret  m  lies in a given set $\left[i,j\right]$. We describe ZKSM according to the representation of Camenisch et al. [37]:(1)$$\mathrm{PK}\left\{\left(m,r\right):Com_{K}=g^{m}h^{r}\left(i\leq m\leq j\right)\right\},$$
where $\;Com_{K}=g^{m}h^{r}\;$ is the secret $\;m\in \left[i,j\right]$ using random values  r. In other words, the above proof will convince the verifier that the secret in the promise $\;Com_{K}$ lies in the set $\left[i,j\right]$.

In general, a zero-knowledge proof system needs to satisfy the following three properties:(1)Completeness. Knowing the validity of the witness statement, the testifier can convince the verifier.(2)Reliability. A malicious prover cannot convince the verifier if the statement is false.(3)Zero-knowledge. The verifier only knows that the statement is correct and does not know the specifics of the statement.

ZKP algorithms include zk-SNARK [38] (concise non-interactive zero-knowledge arguments), ZKBoo [39], zk-STARKs [40], etc. It has been abundantly demonstrated that using ZKP for verification will effectively solve many problems [41,42,43]. ZKP algorithms can be classified into two types of interactive and non-interactive algorithms based on whether there are challenge—response processes between the prover and the verifier. In this paper, we use it to implement identity information hiding and verification in blockchain due to the minimal computational effort of zk-SNARK verification and concise proofs [44].

Go-snark provides a process model for off-chain computation and on-chain verification based on the zk-SNARK algorithm Groth16 [45] protocol. Go-snark contains a constraint generator, a compiler for witnessing and proving, and generators. Specifically, we can perform the operations shown in Figure 1 with the help of go-snark.

The workflow is as follows:

(1)The trusted third party generates a specific domain-specific language (DSL) code in a readable manner, and the code can be compiled to generate a circuit file  cf. 

(2)The circuit file  cf  is converted into a constrained circuit  C  described by the Rank 1 Constraint System (R1CS) [46], which is compatible with the zk-SNARK, where constraint circuit C is an abstraction of the circuit. 

(3)As with other zk-SNARK algorithms, go-snark performs a setup phase to share a common reference string (CRS), which generates two keys, including a proof key  pk  and a verification key  vk. 

(4)Before generating a zk-SNARK proof, the verified program needs to provide the generator with the primary input  pi  and auxiliary inputs  ai  to compute the witness $\;\vec{w}\;$ for zk-SNARK that satisfies the constraints.

(5)The proof π of zk-SNARK is computed based on the witness $\;\vec{w}\;$ and proof key pk. 

(6)The proof  π  combined with the primary input  pi  can be verified by the verification key  vk. The returned authentication result will participate in the system’s identity management.

### 3.2. Architecture and Main Procedures

In this section, we first outline the overall scheme of the system called Health-zkIDM, then describe the entities included in the system, and finally describe the authentication process and algorithm principles in detail.

#### 3.2.1. Program Overview

To solve the user identity privacy problem in authentication, we propose a healthcare identity authentication system called Health-zkIDM based on zk-SNARK and Hyperledger Fabric. By verifying the identity proof information on the chain through zk-SNARK, we solve the possible user privacy disclosure in the authentication process. Specifically, a trusted third party establishes the blockchain and deploys the chaincode on the Fabric. A user and a trusted third party jointly complete the setup step in zk-SNARK to generate the proof of identity information. The user uploads their proof of identity information to Fabric, which triggers chaincode to verify that the information is correct through zk-SNARK. When a user visits a healthcare organization, the user initiates an access request to the healthcare organization. Then, the authorized healthcare provider retrieves the user’s identification information from the blockchain and authenticates the user’s identity. The whole process does not cause privacy disclosure because the proof information does not involve the user’s private information.

#### 3.2.2. Entities

The entities within the Health-zkIDM system include: users (patients and healthcare providers), blockchain and chaincode, healthcare organizations, and trusted third parties. Each entity is briefly described as follows:User: A user who accesses the proposed system can be a patient or a healthcare provider. As the owner of the identity information, the user can conduct the trusted setup of zk-SNARK through the client and upload the generated identity proof information to the system. Each user has a unique ID that represents the user’s digital identity. The user can use the ID to view the information on the blockchain.Healthcare Providers: Healthcare providers are responsible for verifying the identity of patients. When a patient visits a healthcare facility, the healthcare provider can verify the patient’s identity by viewing the data on the blockchain. Since the on-chain data is not related to patient privacy, untrustworthy healthcare providers cannot reveal the patient’s identity.Blockchain and Smart Contracts: The consortium blockchain is used to provide a secure distributed management service for the user’s identification information. Blockchain platforms should support Fabric chaincode. Healthcare providers become verifiers by invoking the chaincode.The trusted third party: The establishment and supervision of the entire system are led by the trusted third party. A trusted third party is required to provide management, authorization, and supervision to each system participant during system operation, such as public security, government, medical regulatory agencies, etc. In addition, the trusted third-party organization is responsible for the trusted settings of the client and zk-SNARK, along with the user.

#### 3.2.3. Existing Authentication Models

Most existing healthcare authentication models [27,33] require a trusted third-party organization to verify user identity attributes and issue certificates to confirm the validity of the information. Users register to the Ethereum using their application. The application includes a secure built-in wallet that contains public and private key pairs. The private key is used to sign authentication and transactions sent to Ethereum. The public key is used to generate accounts on Ethereum. This account is further used to deploy smart contracts on Ethereum. The ID is the address of the smart contract deployed by each entity. The unique ID is used for the identification and authentication process. Figure 2 illustrates the existing Ethereum blockchain-based authentication models in healthcare. Table 2 describes the specific steps for registration and verification of existing identity models. 

#### 3.2.4. Improved Authentication System

Figure 3 shows the framework of the Health-zkIDM system.

First, the user registers the proof of identity information through the client. The client completes four steps: input, compile circuit, trusted setup, and generating proof for the primary input file, trusted setup file, and proof file. Further, the user obtains a unique identification ID and uploads the above files to the Fabric.

When a user visits a healthcare facility, the healthcare provider receives a request for access from the user and needs to ask the user for permission to access the blockchain. 

After the user is authorized, the healthcare provider invokes the identification information on the chain to verify the user’s identity based on the user’s unique identity ID.

Finally, based on the verification result, the healthcare provider decides whether to provide the user with the appropriate service.

In the proposed framework, the Health-zkIDM system is divided into two phases: the registration phase and the invocation phase.

Registration phase: As shown in Figure 4, the system involves zero knowledge and blockchain. During the registration phase, the following steps are required:

Deploy chaincode: Trusted third-party organizations implement chaincode through the client and deploy it to Fabric’s peer nodes.Initialization: The user inputs the polynomial constraint part of the coefficients through the client SDK, and the other part of the coefficients are randomly generated by the client to design the corresponding constraint circuit  C  based on the circuit satisfiability problem. The constraint circuit is described in detail in the function CircuitConstraintSetup. The user inputs the primary input, pi, auxiliary input, ai, obtains the primary input file, and witnesses $\;\vec{w}\;$ by computation. Further, the user uses the function TrustedSetup to generate the trusted setup file ts, which contains the proof key pk and the verification key vk. With the function GenerateProofs, the proof file can be generated using the $\vec{w}\;$ and pk. Since the primary input file, trusted setup file, and proof file do not contain privacy, they can be packaged and uploaded to the Fabric as public data.User registration: The user generates a unique ID through the App, and the unique ID is the user’s digital identity. Function RegisterIdentity describes the user registration process in detail. When uploading data, the blockchain automatically runs the chaincode to check the legitimacy and correctness of the proof information.

Invocation phase (authentication phase): According to Figure 5, the invocation phase requires the healthcare provider to obtain access rights from the user and to authenticate the identity of the user accessing the healthcare organization.

The healthcare provider becomes the verifier by a chaincode deployed on the blockchain. The authentication process takes place in the client (see function Verify). Only healthcare providers allowed by the user can register with Fabric and invoke the chaincode to view the user’s proof of identity information. The verification of the proof information is used to decide whether to provide the service to the patient.

#### 3.2.5. Comparison

In the registration phase, the existing blockchain-based identity authentication model requires the creation of public-private key pairs. The verification of the user’s identity is accomplished through frequent challenge-response operations, i.e., public-private key pairing operations, between the user and trusted third-party organizations. In contrast, the improved model can automatically execute the verification process after uploading the identity proof information, and the open and transparent chaincode ensures the trustworthiness of the verification.

In the authentication phase, the existing authentication model requires the public key registered on the blockchain by a trusted third-party organization to be retrieved based on the ID and the public – private key pairing to verify the user. However, the improved model can automatically verify the information with the zk-SNARK algorithm while the healthcare organization retrieves the information on the chain to verify the user’s identity.

Table 3 shows the differences between the existing models and improved models. This improvement greatly reduces the interaction between users and trusted third-party organizations and between users and healthcare organizations, saves the time of interaction operation, and eliminates the possibility of attackers attacking during the information interaction. By using ZKP, users can completely hide the registration information and the information owned by the user is unknown, achieving effective privacy protection.

#### 3.2.6. Algorithms

In this section, we describe specifically the algorithms mentioned in Section 3.2.4, including the algorithm in the zero-knowledge proof and the algorithm for chaincode.

Algorithm 1 describes the method used in the zero-knowledge proof. The polynomial constraint, which serves as the basis for compiling the circuit file, restricts the user’s input and is an important part of user authentication, as shown in the function CircuitConstraintSetup. Function TrustedSetup describes the user’s ability to generate pk  and  vk  steps in detail. Function GenerateProofs describes the implementation of generating proofs. Function Verify describes the implementation of proof verification.
**Algorithm 1** Zero-knowledge proof function1: **function** CircuitConstraintSetup (c1,c2)2:  Generate a polynomial px, where two coefficients are  c1, c2, and the remaining coefficients are randomly generated by the client.3:   circuit file cf ←px.{Write polynomial px into the circuit file in Golang language}4:   According to the circuit file cf, circuit C is generated by a circuit language called R1CS.5:   return C6: **function** TrustedSetup (C)7:   (A→,B→,C→)←R1CStoQAPC.{call the go-snark’s function R1CStoQAP to generate three vectors A→, B→, C→}8:   ts, pk,vk←keyGen(A→, B→, C→). where ts is a trusted setup file, pk  is a proving key, and  vk is a verification key.{call the go-snark’s function keyGen to generate keys for proof and verification}9:   return pk, vk.10: **function** GenerateProofs (pk,pi,ai,cf,px)11:   Set w→ :=pi, ai, where w→ is witness, calculate by the user’s primary input pi and the user’s auxiliary input ai.12:   compute π := GenerateProof (cf,pk,w→,px).{call the go-snark’s function GenerateProofs to generate proofs π for prover}13:   return π14: **function** Verify (vk, pi, π)15:   compute verified := VerifyProofvk, pi, π
{call the go-snark’s function VerifyProof to verify}16:   **if** verified = true **then**17:    return 1.18:   **else**19:    return 0.20:   **end if**

Chaincode function describes the implementation of user identity registration, reset, query, and revoke operations, as shown in Algorithm 2.
**Algorithm 2** Chaincode function1: **function** RegisterIdentity (id, ts, pi, π)2:   **if** id exist **then**3:    return 0.4:   **else**5:    Fabric ← id, ts, pi,π where id is the user’s ID6:    return 1.7:   **end if**
8: **function** ModifyIdentity(id, ts′, pi′, π′)9:   **if**
id exist **then**10:   m :=Modifyts′, pi′, π′11:   Fabric ←m12:   return 1.13:  **else**14:   return 0.15:  **end if**16: **function** QueryIdentity(id)17:  **if**
id exist **then**18:   Message :=Queryid19:   return Message.20:  **else**21:   return 0.22:  **end if**23: **function** RevokeIdentity(id)24:  **if**
id exist **then**25:   Revokeid26:   return 1.27:  **else**28:   return 0.29:  **end if**

## 4. Evaluation and Analysis

To simulate the application scenario of zk-SNARK and Fabric blockchain-based authentication system, we developed a prototype system including chaincode and a client to evaluate the feasibility and performance of the Health-zkIDM system. The chaincode prototype in the system is written in Golang. The client framework is go-snark + Golang + Html. Client operating environments of three virtual machines is all Ubuntu 20.04 (64-bit), Intel Core i7-9700KF@3.6GHz64G, with one virtual machine deploying the orderer node and the other two virtual machines deploying org1 and org2.

### 4.1. Use Cases

Suppose a user named Alice is visiting a healthcare facility, and the healthcare provider asks Alice to give her proof of identity information and viewing privileges to verify her identity. Since Alice is using the system for the first time and does not have her proof information, Alice needs to register with Health-zkIDM. Alice obtains her unique ID and registers her proof information through the client. The proof of identity is automatically verified once it is uploaded to the chain to ensure the correctness of the data on the chain. Figure 6 shows that Alice has registered her identity information. After the registration is completed, Alice grants permission to the healthcare provider to view the information and transmits her unique ID. After the healthcare provider obtains the ID, it queries Alice’s identity proof information based on the ID and automatically verifies it through the client, and the verification result is returned to the healthcare provider. The medical service provider decides whether to provide services to Alice based on whether the ID is authenticated or not.

### 4.2. Performance Analysis

#### 4.2.1. Storage and Time Consumption

The experiments show that the storage and time consumption of this scheme is mainly in the TrustedSetup, GenerateProofs, and Verify algorithms. Table 4 shows the time used to perform TrustedSetup and generate proofs when the registration is performed by the local client, and Table 5 shows the size of the input file and the output file according to the algorithm during registration, with each result being the average of 100 tests. The time to compute the zk-SNARK proofs depends on the computational resources, code logic, and amount of operations allocated by the proof procedure.

#### 4.2.2. Throughput

Hyperledger Caliper [16] is a blockchain performance benchmarking framework, and we use the caliper to evaluate the proposed scheme. The identity registration, identity modification, identity query, and identity revocation functions of the chaincode are tested by simulating 20,000 transactions, respectively. Success rate, maximum latency, minimum latency, average latency, and throughput are the simulation metrics considered. Table 6 shows the test results of a virtual machine. To avoid affecting the block synchronization speed, we set the block out time interval to 5 s. We observe that the throughput of the system reaches more than 460 TPS, where TPS represents the number of transactions per second. In addition, we also used three virtual machines to test the throughput in the case of multiple machines, as shown in Table 7. It can be found that under the same conditions, the throughput of a multi-virtual machine deployment test is smaller than that of a single virtual machine. However, the throughput of multiple computers still reaches 330 TPS, which may be caused by network interaction delay and computer resource preemption.

In addition, we evaluate the impact of the number of workers on blockchain performance, where workers represent the number of people executing transactions at the same time. Figure 7 shows the relationship between the number of workers and TPS. It can be observed that the TPS gradually decreases when the number of workers becomes larger. To achieve high TPS, a smaller number of workers is required. However, in the real scenario, the number of workers representing patients and healthcare providers is very high [12]. For 100 workers with simultaneous transactions, we observe that TPS can also reach more than 260. Compared to the literature [33], the throughput is improved by 3 to 5 times.

### 4.3. Security Analysis

In Health-zkIDM system, security should be ensured at both the bottom layer and the application layer [11,47,48,49,50]. At the bottom layer, all transactions can be executed correctly; at the application layer, the user’s private information cannot be disclosed. The security constraints we need to analyze are shown in Table 8, which compares the conditions required to implement these security constraints with the purpose for which they are implemented. The prerequisites required to perform a security analysis of the proposed solution are as follows:

Prerequisite 1: The user’s subsidiary input  ai  has been properly stored and not leaking.

Prerequisite 2: All information stored in the local database has been properly stored and not leaked.

#### 4.3.1. Security of the Bottom Layer

Zero-knowledge security: The go-snark tool used in the system is based on the Groth16 protocol of zk-SNARK, whose security has been verified in the literature at [45].

Anti-Replay Attack: The attacker intercepts legitimate patient identity input from a transaction $\left(vk,\;pi,\;\pi \right)$. The attacker can use this identity to log into the healthcare facility and enjoy all the services of that patient. However, in our algorithm, according to the following equation:(2)$$\left({vk}^{\prime },\;{pi}^{\prime },\;\pi^{\prime }\right)=\mathrm{Updata}\left(vk,\;pi,\;\pi \right),$$
whenever a healthcare provider completes a verification, the patient’s identification information $\left(vk,pi,\pi \right)$ will be reset to a new $\left({vk}^{\prime },{pi}^{\prime },\pi^{\prime }\right)$ and there is no secondary use problem. Therefore, the solution prevents replay attacks.

Special computational security: In the zero-knowledge trusted setup phase, we are required to set the constraints together with the trusted third party establishing the blockchain. Function CircuitConstraintSetup ensures the unknown nature of the constraints, making it impossible for both the user and the trusted third party to obtain the complete constraint information. Therefore, an attacker cannot reveal the zero-knowledge by attacking either party to obtain the constraint information.

#### 4.3.2. Security of the Application Layer

User privacy protection: firstly, the user’s identification information does not involve the user’s real private information. Secondly, the user’s identification information is encrypted in one way by zero-knowledge proof, which ensures the security of data on the chain. Most critically, when users access the nodes of healthcare organizations, they need to grant viewing permission to the healthcare organizations, which are to a certain extent trusted to the users, and even if they grant permission to the untrustworthy healthcare organizations, the proof of information data they obtain will not disclose the user’s private information, because the data does not involve the user’s real information. The above security analysis shows that:Healthcare providers can verify the identity of users without obtaining their privacy.Any attacker cannot obtain the user’s privacy by analyzing the interaction between the user and the healthcare provider.The user’s proof information encrypted with zero-knowledge proof is stored on the blockchain and cannot be revealed. It is done so that no local computer, blockchain, healthcare provider, or trusted third party can directly view the user’s privacy. Patients have absolute control over their identity information, and only the actor who grants permission can verify the patient. Therefore, the system protects the privacy and security of users.

## 5. Conclusions

Authentication management technology is key to addressing the security and confidentiality of user privacy. However, the current healthcare industry mostly uses centralized identity management, which still has many security concerns. The development of blockchain technology has provided researchers with more perspectives to design identity management systems to enhance their security. In this paper, we propose a partially decentralized healthcare identity authentication system, Health-zkIDM, based on zero-knowledge proof and blockchain technology, which transfers the control of identity information from the service provider to the end user. The proposed system allows patients and healthcare providers authentication across different healthcare organization domains, facilitating provider – patient interactions without relying on a central service provider. The unique identifier ID is used as the digital identity of the user to identify different users. The user uses the chaincode to store the identity proof information generated by the zero-knowledge proof technology into the Fabric for verification by the healthcare providers.

To evaluate the performance of the proposed system, we implemented the chaincode on Fabric managed by a trusted third party and tested the performance of the system through the caliper. This includes the time consumed by the chaincode execution and the required storage space on the chain. Future work will focus on overcoming the shortcomings of Health-zkIDM as well as on extending the system to make it suitable for a wider range of application scenarios.

## Figures and Tables

**Figure 1 sensors-22-07716-f001:**
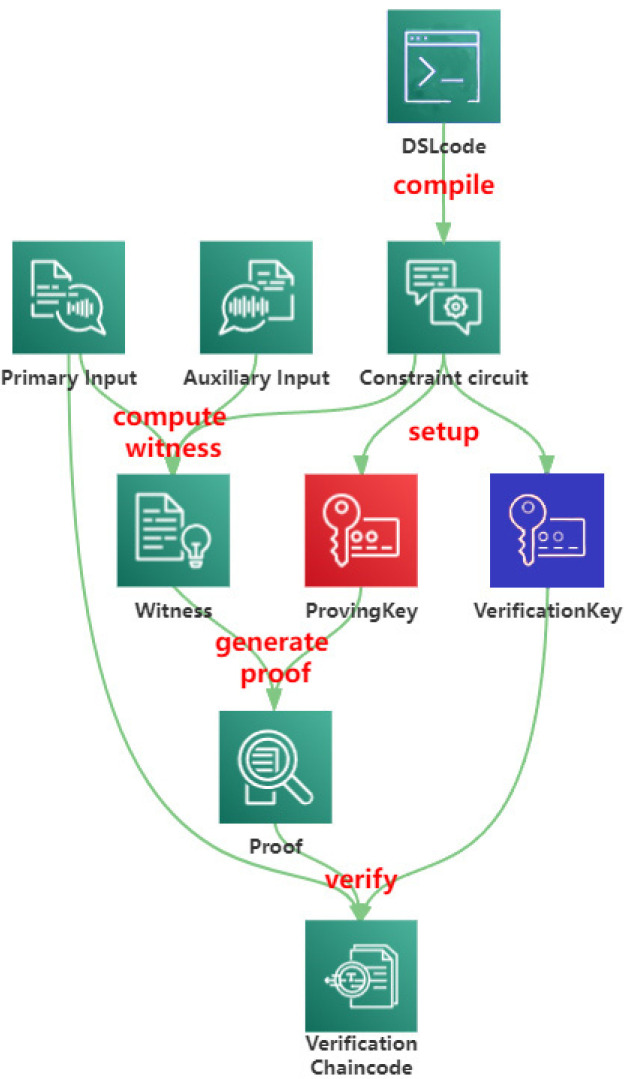
The process of go-snark.

**Figure 2 sensors-22-07716-f002:**
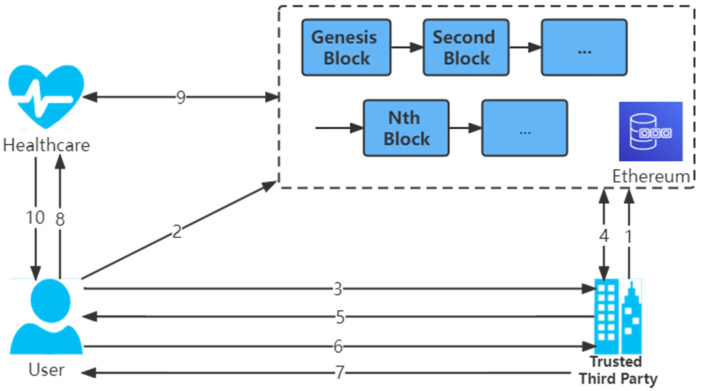
The existing identity model. (1) A trusted third-party organization deploys a smart contract. (2) The user uploads the public key to Ethereum for account generation. (3) The user sends the ID obtained from registration to the trusted third-party organization. (4) A trusted third-party organization requests the public key stored by the user on Ethereum according to the ID. (5) A trusted third-party organization sends the challenge information encrypted by the public key to the user. (6) The user decrypts the challenge information by the private key to ensure that the user is the owner of the ID and the public key. (7) A trusted third-party organization responds to the registration process and uploads the user’s ID and public key to Ethereum and marks it as verified. (8) The user sends the ID to the healthcare organization. (9) The healthcare organization uses the ID to request the user’s public key on Ethereum. (10) The healthcare organization uses the public key to send a query message to the user, and if the user responds correctly, the verification is successful.

**Figure 3 sensors-22-07716-f003:**
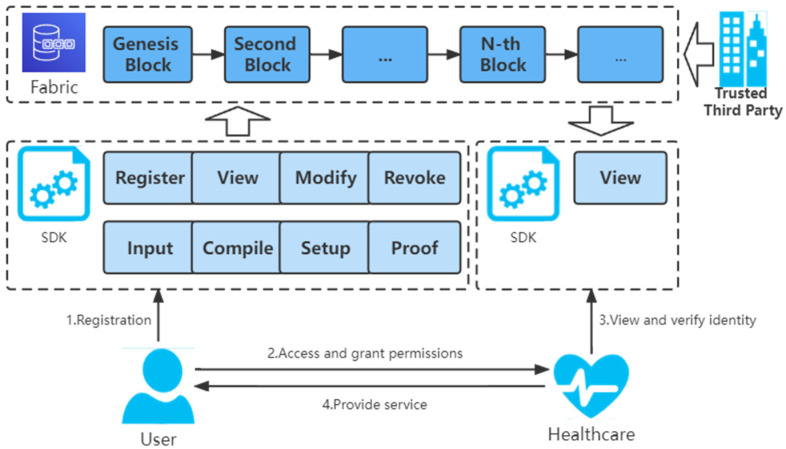
The improved identity model.

**Figure 4 sensors-22-07716-f004:**
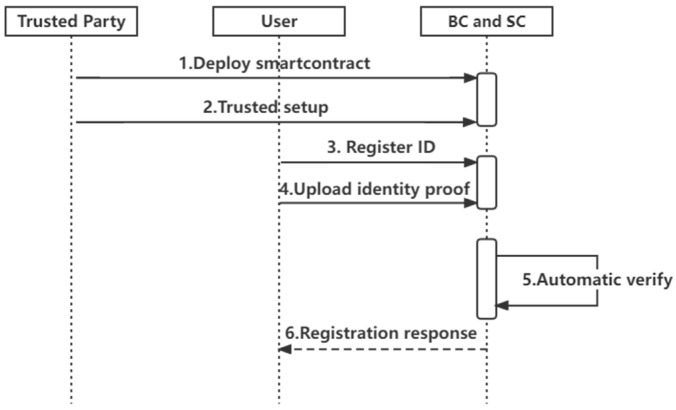
Registration sequence diagram.

**Figure 5 sensors-22-07716-f005:**
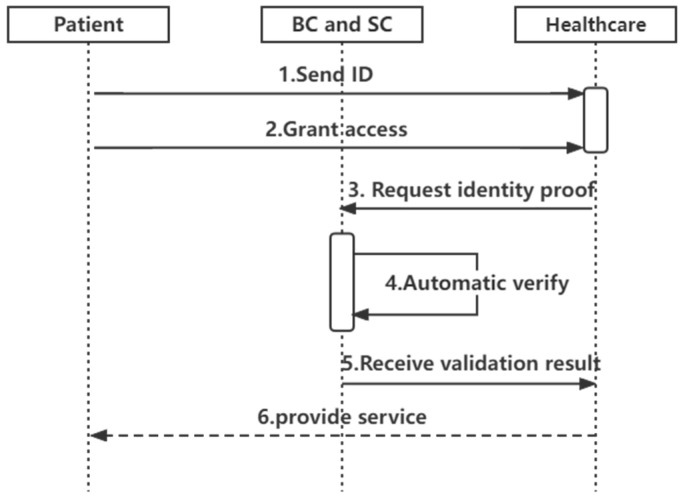
Authentication sequence diagram.

**Figure 6 sensors-22-07716-f006:**
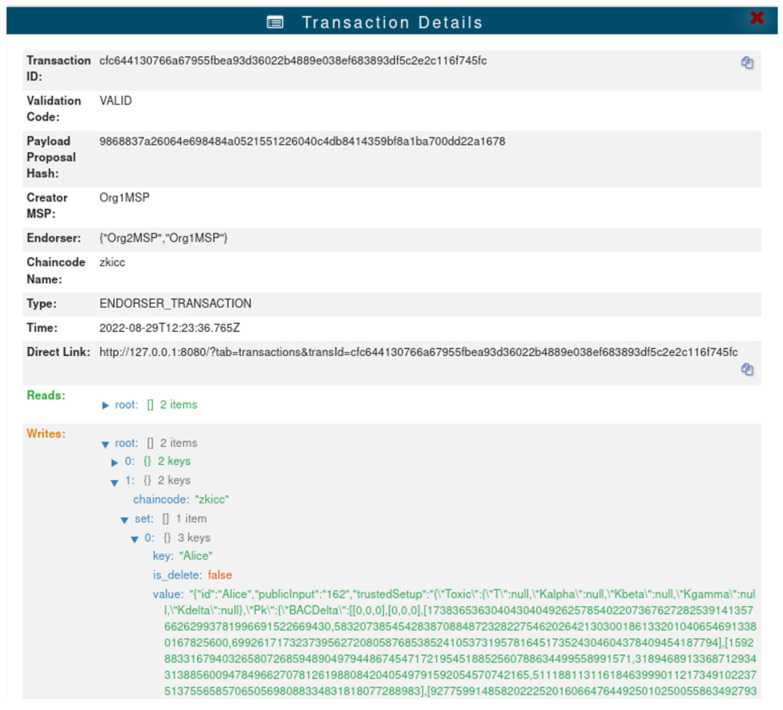
Register identification information.

**Figure 7 sensors-22-07716-f007:**
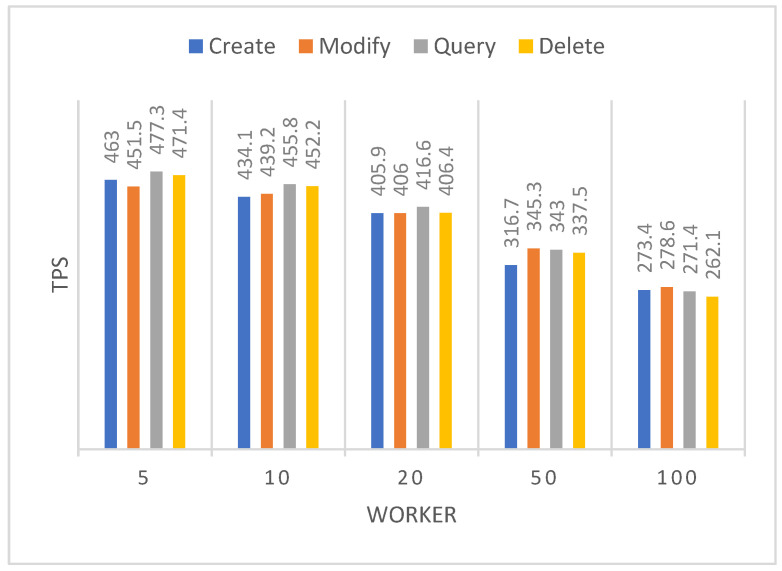
Throughput of different numbers of workers.

**Table 1 sensors-22-07716-t001:** Comparison of related proposals.

Ref	Technology	Solution	ZKP	Healthcare Scenarios	Identity Authentication	The Main Contributions
[19,20]	Hyperledger Indy	Sovrin	√	×	√	Create self-sovereign identities
[23,24]	Bitcoin	ShoCard	×	×	√	Provide a trusted authentication
[27]	Ethereum	MedRec	×	√	√	Manage medical records
[28]	Public blockchain & IoT	HealthChain	×	√	×	Remotely monitor and share health data
[29]	Public blockchain	-	×	√	√	Provides a decentralized authentication method
[30]	Ethereum	BiiMED	×	√	×	Secure sharing of EHR
[31]	Public blockchain & IoMT	BSDMF	×	√	×	Provide secure data management
[32]	Hyperledger Fabric	-	×	√	×	Strengthen privacy protection in data sharing
[33]	Ethereum	Health-ID	×	√	√	Security management of identity information
Our work	Hyperledger Fabric	Health-zkIDM	√	√	√	Provides a secure IDM in healthcare scenarios

**Table 2 sensors-22-07716-t002:** Registration and validation steps for existing models.

Deployment	Step 1	Deploy smart contract
	Step 2	Uploads public key
Registration	Step 3	Send the registered ID
	Step 4	Request public key
	Step 5	Send challenge message
	Step 6	Response to the challenge
	Step 7	Response whether the registration was successful
Verification	Step 8	Send the registered ID
	Step 9	Request public key
	Step 10	Challenge and verify

**Table 3 sensors-22-07716-t003:** Comparison of existing models and improved models.

Comparison	Number of Interactions between Users and Third Parties	Number of Interactions between Users and Healthcare Providers	Storage and Authentication of Public and Private Keys	ZKP	Time of Interaction	Security of the Interaction Process
Existing models	4	4	√	×	Long	Normal
Improved models	0	2	×	√	Short	Strong

**Table 4 sensors-22-07716-t004:** Time consumption for trusted setup and proof generation.

TrustedSetup Time	GenerateProofs Time	Verify Time	Total
711.0457 ms	285.2845 ms	2.1122 s	3.1685 s

**Table 5 sensors-22-07716-t005:** The input file, trusted settings file, and proof file storage consumption (unit: Byte).

Input		TrustedSetup	GenerateProofs
Primary	Auxiliary		
10	5	25,160	962

**Table 6 sensors-22-07716-t006:** Caliper report of Fabric performance under single computer.

Name	Succ	Fail	Send Rate	Max Latency (s)	Min Latency (s)	Avg Latency (s)	Throughput (TPS)
Register identity.	20,000	0	481.8	2.68	0.05	1.07	463.0
Modify identity.	20,000	0	468.4	2.68	0.07	1.11	451.5
Query identity.	20,000	0	496.9	2.57	0.10	1.04	477.3
Revoke identity.	20,000	0	490.8	2.67	0.02	1.09	471.4

**Table 7 sensors-22-07716-t007:** Caliper report of Fabric performance under multiple virtual machines.

Name	Succ	Fail	Send Rate	Max Latency (s)	Min Latency (s)	Avg Latency (s)	Throughput (TPS)
Register identity.	20,000	0	344.7	2.94	0.02	1.29	322.5
Modify identity.	20,000	0	352.4	2.89	0.04	1.27	330.4
Query identity.	20,000	0	367.1	2.61	0.03	1.25	345.0
Revoke identity.	20,000	0	368.6	2.88	0.02	1.26	343.6

**Table 8 sensors-22-07716-t008:** Security constraints of Health-zkIDM.

Security Constraint	Condition	Purpose
Zero-knowledge security	Appropriate bilinear groups should be chosen.	Prevents the input used to generate the proof from being revealed.
Anti-Replay Attack	Used proofs should not be validly validated.	Prevents attackers from replay proofs to access healthcare organizations and enjoy services.
Special computational security	The trusted setup phase, performed by a trusted third party and the user, needs to be verified correctly.	Prevent attackers from revealing zero knowledge by attacking either party to obtain constrained information.
User privacy protection	When the identity is verified, the user does not have to present the real identity information.	Prevent the user’s identity information from being exposed to attackers.

## Data Availability

The data that support the findings of this study are available after the article is accepted and published.

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
