# Peer review of "Health-zkIDM: A Healthcare Identity System Based on Fabric Blockchain and Zero-Knowledge Proof"

_sensors, 2022, doi:10.3390/s22207716_

Round 1

Reviewer 1 Report

1.     A space id is required within the reference. This is a wrong failure[2] and it is a correct failure [2].

2.     In related work you should justify why your approach is different from the previous approaches. Also, provide a table of comparison with the other approaches.

3.     Section 3 should be Labelled as the Methodology section and then include subsections 3.1, 3.2, and 3.3 as a Review (3.1) in the Methodology section. Then section 4 should be 3.2 and so on. You should check the given paper. [ An efficient and reliable core-assisted multicast routing protocol in mobile Ad-Hoc network].

4.     Evaluation and analysis should be section 4 and so on.

5.     5. Result section is not satisfactory. Please include more diagrams and explanations.

6.     Also provide a comparison with some other protocols.

Reviewer 2 Report

Summary: In this article, the authors "propose Health-zkIDM, a decentralized identity authentication system based on zero-knowledge proof and blockchain technology, which allows patients to identify and verify their identities transparently and safely in different health fields, and promotes the interaction between IDM providers and patients."

  Comments and Suggestions: 1. Authors need to improve the whole writing quality of the paper. 2. English and Punctuation mistakes have to be avoided (proofreading is needed) 3. In the abstract, the authors say "Health-zkIDM system can achieve throughputs higher than 400 in Caliper". It is unclear which measure unit is used to express throughputs (400 what?). 4. In the introduction, contribution 3 may be divided into two parts. 5. The authors need to insert a table that summarizes the main findings of the related work section. 6. The two following references may be added to the related work section: - Blockchain-assisted secured data management framework for health information analysis based on Internet of Medical Things - Blockchain technology for healthcare: Enhancing shared electronic health record interoperability and integrity 7. In the related section work, the authors may give a brief overview of other fields (other than healthcare) in which the proposed technique is used. 8. The title of section 3 contains some Asian words. 9. Line 150: "Permission blockchain can be further divided into a consortium". Do you mean "permissionless blockchain" or "permissioned blockchain" 10. Section 3.2 contains several formal entities and equations which make the section hard to understand. 11. A more significant tittle for Section 3.3 may be proposed. 12. In section 4.3: "Fig.2 illustrates the existing Ethereum blockchain-based authentication models in healthcare." ==> please add references about existing models. 13. The ten steps presented on Page 10  can be inserted into a table for making them easier to read. 14.  The comparison proposed in Table 1 is too simplistic and does not convince readers about the quality and efficiency of the proposed approach. More details may be added. 15. Algorithms presented in Section 4.6 may be better merged into only one or two algorithms. Indeed, the presented algorithms are too short and too simplistic. 16. In section 5, it is mentioned that only one computer was used for experimentation. If this is true then the experimentation seems to be very limited. 17. The use case presented in section  5.1 can be introduced earlier in the paper in order to help readers understand the goal of the proposed approach. 18.  The security analysis presented in Section is too limited and too simplistic and it is not clear how it may really guarantee the robustness of the proposed approach against possible attacks. 19. The authors need to explain how the many challenges related to the use of blockchain technology can be solved in their work. 20. The authors need to share a video recording of their experimental work and also share details about how the experimentation can be reconducted by interested readers.     

Reviewer 3 Report

The paper proposes an interesting approach to handling identity management using blockchain-based technology, using zero-knowledge proofs. A prototype was developed and tested in Hyperledger Fabric. The problem posed by the paper is interesting and certainly of use to healthcare environments.

*** Technical inaccuracies and errors ***

There are some significant technical errors in the paper, which in general recommend a reject from my side.

Some examples below:

[*] In the Introduction, authors claim: "Blockchain is decentralized [...] and transparent". This is incorrect. A blockchain is not decentralized by inherent nature - in fact, a "blockchain" is a ledger of data. Whether it is distributed or not, that is a completely different story - the authors, in this sentence, assume the blockchain ledger is distributed (something that is not specified upfront!), and furthermore, that the access permissions to the distributed ledger itself are not required (e.g. "transparent"), again something that is not true. Since authors used Hyperledger to run their experiments, they should be most aware of the fact that most Hyperledger-based solutions are, actually, permissioned distributed ledgers - meaning, permission is required for accessing the data in the ledger.

What I recommend in these cases, is for the authors to go back and understand the basics of what blockchain really is: a) what is an actual blockchain, b) how the technology ties in with distributed systems, c) what consensus mechanisms are, d) what types of distributed ledgers using blockchain technology are out there (permissioned/permissionless, private/public, etc.), and then rewrite this accordingly. As it is now, it is incorrect, and simply inherently portrays the lack of knowledge about blockchain technology per se.

[*] In the Introduction, authors claim: "Zero-knowledge proof(ZKP) is an authentication protocol that can prove that a statement is correct without revealing any useful information". This is again, incomplete/incorrect. Define "useful information" in this context, and why "revealing it" is the key of this protocol. In other words, useful to whom, and why?

[*] In Introduction, the concept of "viz. user" is presented, but not introduced. The reader does not know what this refers to. At the same time, in the same paragraph, they claim "A user is a collection of patients and healthcare providers", however it is not clear what the "user" refers to - an user in the blockchain? An user in the paper's proposed ecosystem?

[*] In Related Work, authors claim: "Identities under this scheme can only be used in specific trusted domains and cannot be authenticated across other domains.", but this is simply not true. Centralized identities do provide authentication services to third parties, whether through publicly (or privately) exposed APIs, or otherwise.

[*] In Related Work, authors claim: "The emergence of blockchain gives researchers a new perspective in decentralized IDM. User identities are no longer under the control of a single institution, thus reducing identity privacy breaches.". This is incorrect technically. A blockchain is a ledger that cryptographically secures connections between blocks of operations, nothing more. The "decentralization" aspect that the authors claim here, is actually referring to a completely different technology - that of distributed systems. And here, we have consensus mechanisms that are in place, to ensure mutual agreement of all parties involved in the consensus process. This makes it obvious that the authors may be lacking a fundamental understanding of how these concepts tie in together. Yes, if you look online in any non-reputable website, a "blockchain" is referred to as Bitcoin or Ethereum - but it is NOT. Blockchain technology had existed 20 years prior to Bitcoin even emerging! It's time to stop this immense flow of misinformation, and misrepresent the technology to the casual readers. Authors need to get their facts straight therefore.

[*] In the Preliminaries, authors claim: "As a kind of distributed ledger, blockchain possesses the characteristics of tamper-resistant and decentralized", which is incorrect. First of all, there is nothing that says a blockchain has to be a distributed ledger, even less "a kind of distributed leger". Second, a blockchain's nature by default, inherently, is to be a ledger of information without any type of relationship to distributed systems, therefore a blockchain cannot be "decentralized" in scientific terms - instead, a "distributed ledger making use of blockchain technology could be decentralized" is a totally different concept (and the correct explanation of the situation as well)!

[*] In the Preliminaries, authors claim: "Due to the huge volume of data" - why is this assumed? Who has assumed there is a huge volume of data? Because it seems like a prejudice to assume that all blockchains imply a "huge volme of data", yet not even define that "huge volume" with actual numbers (e.g. what makes it "huge"?).

[*] In the Preliminaries, authors claim: " it makes the transaction extremely inefficient and difficult to be applied to other scenarios except for cryptocurrency and financial fields". This is technically incorrect. The scalability of a distributed blockchain-based ledger has absolutely nothing to do with its size!

[*] In the Preliminaries, authors claim: "As distributed programs running on the blockchain, smart contracts automatically execute a set of predefined rules to complete transactions." Again, technically incorrect. Smart-contracts are triggered by various events, and do not "automatically execute". Furthermore, there is no "set of predefined rules", but rather the "code/bytecode" that developers write and compile for that specific smart-contract!

*** Fundamental flaws ***

The authors claim that their solution is decentralized (which, generally, refers to the consensus mechanism used in distributed systems), yet their proposed architecture used a "trusted third party". This is not decentralization per se in distributed systems, especially since the authors claim that the trusted third party is the one hosting the blockchain and deploying the chaincode! This needs to be rewritten, and corrected, as it is a fundamental flaw in the definition and comprehension of basic concepts that most of the public (e.g. intended audience) is not familiar with, including: a) blockchain technology, b) distributed systems, c) consensus in distributed systems, d) permissioned/permissionless access to ledger, e) private/public access to the blockchain, etc.

*** Comprehension issues ***

Some ideas could be better structured, and their idea should be rephrased, so it makes more sense. Some examples below:

[*] In the Introduction, authors claim "When patients visit the institution, some operations such as information recovering and repeated registration are inevitable". While this to me makes more sense when applied to other institutions (on which the user had not signed up for upfront), it makes no sense if intended for the same institution (assuming he had signed up for it a-priori). So I believe this needs clarification.

[*] In the Introduction, authors claim: "The trusted setup for zero-knowledge proofs is generated by trusted third parties with using users information, which effectively prevents attackers from attacking third party leading to information leakage problems". This is a confusing statement. How is the trusted setup for ZKP "effectively" preventing attackers from attacking the third party? It is not clear at all, not even technically. Therefore, needs further clarification.

[*] In Introduction, authors claim: "The results show that our system can reach a throughput of over 400." - what is the throughput measured in? Operations per second? Megabytes per second? Operations per block? etc. This is unclear, and requires further clarification.

[*] In Introduction, authors claim: "Through security analysis, the system is safe and reliable at the bottom layer and the application layer" -> what "security analysis" is that? What criteria has been used (and classified as "security analysis"), that concluded the system is "safe and reliable"? Needs further clarification.

[*] In section 3.2, authors state "...to calculate the value of the hidden secret...", but it is redundant. I assume the authors wanted to emphasize "the value of the secret key". Overall, the entire sentence: "The cryptographic commitment scheme[31] allows the prover to calculate the value of the hidden secret to ensure that no one can know that the value corresponds to the hidden secret." needs rephrasing, as it is hard to follow and ambiguous.

[*] In section 3.2, there needs to be a space between "value" and "of any", in: "...the secret valuem of any information". Also here, the phrase "Given a commitment value ? , it is difficult to exist a commitment value" needs to be rephrased, as it makes no sense. What does "difficult" mean in this context?

[*] In section 3.3, authors state " It has been abundantly demonstrated that using ZKP for verification will effectively solve many problems", but provide no references of any kind from the bibliography to support this statement. Also, the sentence "ZKP algorithms can be classified into two types of interactive and non-interactive algorithms based on the presence or absence of challenge-response interactions." needs to be rephrased/rewritten, as it hard to read.

[*] In section 3.3, authors introduce the acronym "DSL", but never explain what it is, or what it refers to. This is also present in Figure 1. This needs to be resolved. Even worse, the workflow on page 5 refers to a "constrained circuit", and it is unclear what this is.

[*] In section 4, authors claim "The user uploads their proof of identity information to Fabric and automatically verifies that the information is correct", but it is not clear how the information is automatically verified to be correct. This needs an explanation.

[*] In section 4, authors state: "Fig.2 illustrates the existing Ethereum blockchain-based authentication models in healthcare.", however there are no references, and there is no logic deduction of how this diagram was built (or what it was based on). This needs further clarification. In addition, technically speaking, I do not yet understand why step 2 in this diagram is the uploading of the public key of the user (this needs clarification, as to why it is relevant and where it is used), and what step 3 ("Registration ID") refers to. Step 4 also says "Request public key", but the public key of whom, and why? Same goes for steps 8 through 10. In addition, I would like to understand how step 9 is relevant, when the healthcare organization uses the ID (what exactly does this mean?) of the user to "query" the user's public key on the Ethereum blockchain? This makes little sense to me. Also important, authors need to clarify whether the user does upload any type of personal information at all to the public blockchain Ethereum in this diagram, and if so, how that information is protected from third parties.

[*] In section 5.2.2, authors claim: "However, in the real scenario, the number of workers initiating transactions at the same time is very high.", but they do not provide sufficient data or information to support this claim. Who are the actual workers? What type of transactions are there initiated? Does it matter what the type of these transactions is (e.g. whether they are relevant to the problem at hand, or not)? These are all questions left unanswered.

[*] In section 5.3.2, authors state: "Most critically, when users access the nodes of healthcare organizations, they need to grant viewing permission to the healthcare organizations, which are to a certain extent trusted to the users, and even if they grant permission to the untrustworthy healthcare organizations, the proof information data they obtain will not disclose the users' private information." I would like to see further clarification on how granting permission to an untrustworthy healthcare organization/actor does not compromise the user's private data. Later on, authors state: "The user's proof information encrypted with zero-knowledge proof is stored on the blockchain and cannot be cracked. It is done so that no local computer, blockchain, healthcare provider or trusted third party can directly access user privacy" - and again, if these actors do not get access, who does get it, and why?

[*] In section 6, authors claim "In this paper, we propose a decentralized healthcare identity authentication system...", but the simple fact that amongst the actors in the architecture we find "trusted third-parties", contradicts the "decentralization" that the authors claim. Further clarification is required as to what "decentralized" really and truly means in this context.

*** Grammar issues & typos ***

The paper has multiple grammar issues and typos. Some of them are outlined below, in no particular order:

[*] In "Introduction": "and the information can not interoperable between them" -> "can not be interoperable between them"

[*] In Introduction: "A user" -> "An user"

[*] In Introduction: "with using users information" -> "with using users' information"

[*] In Introduction: "The chaincode is deployed on the Fabric and are used to upload and verify user identity information" -> "...and is used to..."

[*] "Laency" -> "Latency" (Table 4)

[*] ...and many, many others throughout the paper. Too many to put here in this rather short review.

Due to these, a thorough proofread would be required, and a grammar check is imperative.

*** Overall recommendation ***

The idea of the paper is interesting, and I can see how this could very well relate to healthcare systems. Given the fact that the experimental results show promising outcomes with respect to previous work in other papers, it does seem as if the paper does improve the state of art.

Nonetheless, there are multiple technical and factual inaccuracies. These aspects, corroborated with the lack of clarity in several key sentences/phrases throughout the paper, make the paper a difficult read, and a somewhat contradictory read just as well (there are places where authors contradict themselves from a technical standpoint, which have been outlined in the review above).

Based on all the above, I feel that the paper is not ready to be published in its current form. After a series of revisions in accordance to the suggestions above, and an improved clarity of thought and of the English language, I believe the paper could be improved (and the factual/technical errors corrected), up to a state where would be comprehensible even by a non-technical auditorium.

Round 2

Reviewer 1 Report

All the comments are addressed

Author Response

Thanks again for your comments and suggestion concerning our submitted manuscript #sensors-1923349 entitled “Health-zkIDM: a healthcare identity system based on Fabric blockchain and zero-knowledge proof”.

Reviewer 2 Report

The authors took into consideration all my remarks and suggestions. The paper may be accepted for publication.

Author Response

(The authors gave the same response as above.)

Reviewer 3 Report

It's good to see some steps were made in clarifying a lot of the aspects that were previously unclear, in my opinion. Some good amount of content had been added, as requested, and that is refreshing to see. A lot of the grammar issues and typos present in the early version, were fixed/resolved also.

Nonetheless, after a careful review of the paper, I still feel there are some issues left unaddressed, and therefore I will be pointing them out as follows.

[***] Technical inaccuracies [***]

[*] In Introduction, authors claim: "The distributed ledger making use of blockchain technology is decentralized, and its transaction process is tamper-resistant and transparent [7-10]", but this is not true again. Blockchain technology has nothing to do with the storage mechanism (e.g. distributed system, or the "transparency" aspect) or with the consensus mechanism (e.g. the "decentralization" that authors are referring to). The three should be completely separated and spelled out, per se, in here. If they would have said "...is OFTEN decentralized...", that would have been more acceptable.

This to me shows there is still confusion as to what really blockchain technology refers to, so I am compelled to recommend further study on the three concepts above. A correct factual and technical representation of these concepts is expected in blockchain-related scientific papers - again, this is not a Bitcoin magazine, to just promote false information to the general public for the sole, financial gain - this is a scientific paper, that needs to stay fair and true to the technical concepts, solutions and capabilities of the information exposed.

After this sentence, it is necessary to remove the "Therefore, ..." and simply say "A trusted distributed authentication system can be created in an untrusted environment...".

[*] In section 3.1.1, authors state: "A distributed ledger making use of blockchain technology possesses the characteristics of being tamper-resistant and decentralized." I agree with tamper-resistant, but not all distributed ledgers are decentralized (e.g. this assumes a specific type of consensus mechanism, which is not implied, not correct therefore) - so this needs to be rephrased.

[*] In section 3.1.1, another incorrect statement is: "Permissionless blockchain, also known as public blockchain, allows any peer to join the blockchain network" - this is not true. Public/private blockchains refer to the level of exposure of the ledger information to the general public, while permissioned/permissionless refers to who can update the data in the ledger - so the two are completely separate.

[*] Also another incorrect statement in section 3.1.1: "Due to slow transaction speed and low throughput, the public blockchain is difficult to be used in high-throughput scenarios other than cryptocurrency and finance" - it is incorrect to assume slow transaction speeds and low throughput of public blockchains, when TRON for example has a good-enough throughput (and others like it, e.g. Solana). Again, do NOT mix the type of blockchain with the consensus mechanism (a decentralized consensus usually creates low throughputs). These are KEY aspects, which are not portraying a correct picture of this technology.

Also here it is incorrect to assume this: “Permissioned blockchain can be further divided into the consortium blockchain and the private blockchain.“. In general, the classification of permissioned/permissionless blockchains has nothing to do with them being public or private, I hopefully clarified that above. Do not mix the two please. A few other sentences in this paragraph need to be rethink as a result thereof.

[*] In section 3.1.1, authors claim “In this paper, we adopt the most popular consortium blockchain, Hyperleger Fabric”, but Fabric can be used as a private chain, public chain, and even permissioned and permissionless. Also here, authors state: “As distributed programs run on the blockchain, smart contracts are triggered by various events during transactions.“, but blockchains contain other types of operations also, not just transactions (transactions are one particular case of a “blockchain operation”). It is these operations that are cumulated in blocks.

[*] Section 3.1.2 mentions this: “Given a secret value "m" and calculating its commitment value "c = Comm_K(k,r)", it is impossible to exist a commitment value "c' = c = Comm_K(k,r)" that is the same as "c" This property ensures that different secrets cannot generate the same commitment.” - by definition, both c and c’ are equal to the same function/algorithm with the same parameters - is there anything that differentiates them therefore, that would clarify this?

[*] In 4.3.1, authors claim: “The user’s proof information encrypted with zero-knowledge proof is stored on the blockchain and cannot be cracked” - remove the word “cracked” and rephrase. Do not use “cannot be cracked” - you have no way of knowing this for a fact - we only know that so far, this is generally considered to be a safe approach, and that there is no known vulnerability discovered for it. Therefore, this needs rephrasing.

[*] Table 8 - do not use the word "cracked" here either. It's not a scientific term for identifying vulnerabilities.

[***] Typos and grammar issues [***]

[*] Minor typo: "A user" -> "An user"

[*] Minor typo: ” a secure inbuilt wallet” -> “a secure built-in wallet”

I would definitely also like to see the key technical inaccuracies resolved, before the paper is accepted and published. However, I believe this is a minor revision at this point, so I will classify it as such.
